# Establishment of Pancreatobiliary Cancer Zebrafish Avatars for Chemotherapy Screening

**DOI:** 10.3390/cells10082077

**Published:** 2021-08-13

**Authors:** Mariana Tavares Barroso, Bruna Costa, Cátia Rebelo de Almeida, Mireia Castillo Martin, Nuno Couto, Tânia Carvalho, Rita Fior

**Affiliations:** 1Champalimaud Centre for the Unknown, Champalimaud Foundation, 1400-038 Lisbon, Portugal; mariana.barroso@research.fchampalimaud.org (M.T.B.); bruna.costa@research.fchampalimaud.org (B.C.); catia.almeida@research.fchampalimaud.org (C.R.d.A.); mireia.castillo@fundacaochampalimaud.pt (M.C.M.); nuno.couto@fundacaochampalimaud.pt (N.C.); tania.carvalho@research.fchampalimaud.org (T.C.); 2Service of Pathology, Champalimaud Clinical Center, Champalimaud Foundation, 1400-038 Lisbon, Portugal; 3Digestive Unit, Champalimaud Clinical Center, Champalimaud Foundation, 1400-038 Lisbon, Portugal

**Keywords:** pancreatic cancer, ampullary tumors, zebrafish xenografts, chemotherapy, personalized medicine, zAvatars

## Abstract

Background: Cancers of the pancreas and biliary tree remain one of the most aggressive oncological malignancies, with most patients relying on systemic chemotherapy. However, effective biomarkers to predict the best therapy option for each patient are still lacking. In this context, an assay able to evaluate individual responses prior to treatment would be of great value for clinical decisions. Here we aimed to develop such a model using zebrafish xenografts to directly challenge pancreatic cancer cells to the available chemotherapies. Methods: Zebrafish xenografts were generated from a Panc-1 cell line to optimize the pancreatic setting. Pancreatic surgical resected samples, without in vitro expansion, were used to establish zebrafish patient-derived xenografts (zAvatars). Upon chemotherapy exposure, zAvatars were analyzed by single-cell confocal microscopy. Results: We show that Panc-1 zebrafish xenografts are able to reveal tumor responses to both FOLFIRINOX and gemcitabine plus nanoparticle albumin-bound (nab)-paclitaxel in just 4 days. Moreover, we established pancreatic and ampullary zAvatars with patient-derived tumors representative of different histological types. Conclusion: Altogether, we provide a short report showing the feasibility of generating and analyzing with single-cell resolution zAvatars from pancreatic and ampullary cancers, with potential use for future preclinical studies and personalized treatment.

## 1. Introduction

Pancreatic cancer (PC) is one of the most lethal solid tumors, with a devastating 5-year overall survival (OS) of only 7% [1,2]. PC is associated with an extremely poor prognosis for several reasons: (1) it is frequently diagnosed at advanced stages, which is often due to lack of symptoms in the early stages of the disease; (2) lack of validated screening programs for early diagnosis, and of precision treatments; (3) it metastasizes microscopically early on in the course of the disease; and (4) a desmoplastic tumor microenvironment (TME) which contributes to low immune infiltration and drug resistance [3,4,5,6,7].

Pancreatic ductal adenocarcinoma (PDAC) is the most common histologic type of pancreatic cancer and accounts for 85–95% of all solid pancreatic tumors [8]. Treatment guidelines for advanced PC include two main options: FOLFIRINOX (folinic acid + 5-fluorouracil(FU) + irinotecan + oxaliplatin) or gemcitabine plus nanoparticle albumin-bound paclitaxel (GnP) [8,9].

However, other malignances can develop in the area of the head of the pancreas due to their intimate anatomical location, such as carcinomas of the ampulla of Vater and distal cholangiocarcinomas, therefore having a similar therapeutic approach [10,11,12].

Ampullary adenocarcinomas (AAC) are rare malignancies which arise from the ampulla of Vater, distal to the confluence of the common bile and pancreatic duct [10]. There are two main distinct histologic sub-types of ampullary adenocarcinoma based on their origin: intestinal and pancreatobiliary [11]. Patients with intestinal phenotype tumors are treated preferentially with fluoropirimidines +/− oxaliplatin, usually FOLFOX (5FU + oxaplatin), whereas biliopancreatic phenotype tumors are usually treated with gemcitabine-based chemotherapy [11].

Distal cholangiocarcinomas, developed in the distal common bile duct, are periampullary neoplasms that may be difficult to distinguish from PDAC and AAC, and, due to their anatomic location, can also be surgically treated by pancreaticoduodenectomy. The use of adjuvant treatment with capecitabine is usually discussed with these patients [12].

Unfortunately, there are still no reliable predictive biomarkers implemented in the clinic that can predict which chemotherapy drug or regimen can provide more benefit for each individual patient. Consequently, many patients can be exposed to unnecessary severe side effects and miss “therapeutical time”. Thus, assays that could directly challenge tumor cells from each patient to the different types of therapy in a few days could have a potential application for guiding individual treatment decisions.

Currently, mouse patient-derived xenograft (PDX) is the most widely used and validated model to predict response to therapy. However, tumor establishment and evaluation of therapy options can take months [13]. Therefore, the mouse PDX model is not feasible for first clinical decision-making.Organoid cultures from patient-derived cancer tissues have become a highly attractive tool to be used as an in vitro screening platform, with very promising results for several tumors, including in pancreatic cancer [14,15]. Organoids were shown to maintain the overall genetic characteristics of the original tissue [14]. Nonetheless, these models still lack many complex interactions observed in a living organism and do not allow, for instance, the evaluation of crucial hallmarks of cancer, such as metastatic or angiogenic potentials.

In the last years, we have been developing the zebrafish PDX “zAvatars” for personalized medicine to help determine the best therapeutic option for each cancer patient [16,17,18,19]. This assay relies on the injection of fluorescently labeled tumor cells into 2 days postfertilization (dpf) zebrafish embryos and accessing tumor behavior and response to anticancer therapy after 4 days. zAvatars offer a short-time assay, single-cell resolution, large numbers of xenografts, and in vivo evaluation of crucial cancer hallmarks, such as proliferation, metastasis, and angiogenesis [13,16,17,18,19]. In our first study, we screened the colorectal cancer (CRC) therapy guidelines—from first- to third-line treatments, evaluating drug efficacy [16]. As a proof of concept, we showed that zAvatars could predict response to CRC adjuvant treatment in 84% of the cases [16]. Recently, we also developed the model to assess sensitivity to radiotherapy [17] and targeted therapies, such as anti-EGFR (e.g., cetuximab) [16], anti-VEGF (e.g., bevacizumab) [18], and PARPi (e.g., olaparib) [20].

Here we aimed to optimize the pancreatic zebrafish Avatar protocol. We started by using a human PC cell line (Panc-1) to screen the treatment guidelines for advanced PC. By analyzing mitotic index, cellular apoptosis, and tumor size, we were able to detect anti-tumor responses with single-cell resolution in just 4 days. As a proof of principle, we also generated zAvatars from pancreatobiliary cancer patients with different histological types. Overall, our results show that zAvatars may constitute a promising in vivo personalized model to screen therapeutic options in tumors developed in the pancreatic region.

## 2. Materials and Methods

### 2.1. Animal Care and Handling

In vivo experiments were performed in the zebrafish model (*Danio rerio*), which was maintained and handled in accordance with European Animal Welfare Legislation, European Guidelines (2010/63/EU), and Champalimaud Fish Platform Program. The study and procedures were approved by the Ethical Committee and Portuguese institutional organizations: Animal Welfare and Ethics Body (ORBEA) and Directorate General for Food and Veterinary (DGAV).

### 2.2. Zebrafish Lines

Experiments were performed using transparent nacre, which has a complete lack of melanocytes due to a mutation in the gene encoding the *mitfa* gene [21], and *Tg(Fli1:eGFP)*, which allows the visualization of blood and lymphatic vessels, through the expression of eGFP linked to *fli1* (endothelial marker) promoter [22].

### 2.3. Patient Samples

Surgically resected samples were collected in rich media, containing a mixture of antibiotics and antifungals, and cryopreserved until injection. When defrosted, tumor tissues were dissociated in Mix1 with EDTA (Sigma-Aldrich, St. Louis, MO, USA), DNase (ThermoFisher, Paisley, UK), and liberase (Roche, Basel, Switzerland) for approximately 20 min at 37 °C (Mix2, Appendix A). For cell labeling, tumor cells were incubated with the fluorescent cell tracker Deep Red (Life Technologies, Carlsbad, CA, USA) for 10 min at 37 °C. Cell suspension was filtered through a 70 mm cell strainer, centrifuged (250× *g*, 4 min), and resuspended in Mix1. Tumor cells were checked for viability with Trypan Blue (Sigma-Aldrich) dye exclusion. A small aliquot of the processed/dissociated tumor sample was used to generate a cell smear, and stained with May-Grunwald–Giemsa (Bio-Optica, Milan, Italy) according to the manufacturer’s instructions for cytological analysis.

### 2.4. Human Pancreatic Cancer Cell Line

Panc-1 was kindly provided by Valérie Paradis at Beaujon Hospital (Clichy, France). This cell line was authenticated through short tandem repeat (STR) profile analysis and tested for mycoplasma contamination.

### 2.5. Cell Culture

Panc-1 was adherently cultured and expanded to 70–80% confluence using Dulbecco’s modified Eagle medium (high glucose) (Biowest, Nuaillé, France) supplemented with 10% fetal bovine serum (Sigma-Aldrich, St. Louis, MO, USA) and 1% penicillin–streptomycin 10,000 U/mL (Hyclone, Marlborough, MA, USA). Cells were maintained with a humidified atmosphere at 5% CO_2_ and 37 °C.

### 2.6. Cell Staining

Cells were labeled with lipophilic dyes: vybrant CM-DiI (ThermoFisher Scientific) at a concentration of 4 µL/mL or Deep Red (CellTracker^TM^, ThermoFisher Scientific) at a concentration of 1 µL/mL. Staining was performed according to the manufacturer’s instructions. Cells were resuspended to a final concentration of 0.25 × 10^6^ cells/µL.

### 2.7. Zebrafish Xenograft Microinjection

Labeled cancer cells were microinjected using borosilicate glass capillaries under a fluorescence scope (Zeiss Axio Zomm. V16) with a mechanical pneumatic injector attached (Pneumatic Pico pump PV820, World Precision Instruments). Cells were injected into the perivitelline space (PVS) of 2 dpf zebrafish embryos, previously anesthetized with Tricaine 1X (Sigma-Aldrich). After injection, zebrafish xenografts remained for ~10 min in Tricaine 1X and then transferred to E3 medium and kept at 34 °C. At 1 day post-injection (dpi), zebrafish xenografts were screened regarding the presence or absence of a tumoral mass. Xenografts with severe edema, cells in the yolk sac, cell debris, or noninjected zebrafish embryos were discarded, whereas successful ones were grouped according to their tumor size, which was classified by comparison with the eye’s size. At 4 dpi, xenografts were sacrificed, fixed with 4% formaldehyde (Thermo Scientific) at 4 °C overnight, and preserved at −20 °C in 100% methanol (VWR Chemicals, Radnor, PA, USA).

### 2.8. Xenografts Drug Administration

At 1 dpi, zebrafish xenografts with similar tumor size were randomly distributed in the treatment groups: control E3 medium, FOLFIRINOX in E3 (4.2 mM 5-FU, 0.18 mM folinic acid, 0.08 mM irinotecan, 0.08 mM oxaliplatin), GnP in E3 (1.6 mM gemcitabine, 365 ng/mL nab-paclitaxel), or FOLFOX in E3 (4.2 mM 5-FU, 0.18 mM folinic acid, 0.08 mM oxaliplatin) for three consecutive days, replaced daily (Appendix A). Zebrafish maximum tolerated concentration was determined using the maximum patient’s plasma concentration of each compound as a reference [23,24,25,26,27,28].

### 2.9. Xenograft Whole-Mount Immunofluorescence

Whole-mount immunofluorescence was performed, starting with a rehydration process through methanol series (75% > 50% > 25%). Next, xenografts were permeabilized with 0.1% triton in PBS and blocked with a PBS 1X, 0.01 g/mL BSA, 1% vol DMSO, 1% triton, and 0.0225% vol goat serum for 1 h at room temperature. Subsequently, xenografts were incubated with primary antibodies diluted at 1:100 overnight. The primary antibodies used were: rabbit anti-cleaved caspase-3 (Cell Signaling Technology, Danvers, MA, USA, code#9661), mouse anti-human mitochondria (Merck Millipore, Burlington, MA, USA, cat#MAB1273), and rabbit anti-phosphohistone H3 (Merck Millipore, cat#06-570). On the following day, xenografts were washed and incubated overnight with secondary antibodies at 1:400, and 50 µg/mL DAPI (Sigma-Aldrich) for nuclei counterstaining. After washing and fixation steps were performed, xenografts were mounted between two coverslips using Mowiol mounting media (Sigma-Aldrich).

### 2.10. Xenograft Imaging and Quantification

All images were obtained using a Zeiss LSM 710 fluorescence confocal microscope, generally with a 5 μM interval using the z-stack function. Generated images were analyzed using FIJI/ImageJ software. We also used the stitching plugin for the whole zAvatar image [29].

To assess tumor size, three representative slices of the tumor, from the top (Zfirst), middle (Zmiddle), and bottom (Zlast), per xenograft were analyzed, and a proxy of total cell number of the entire tumor (DAPI nuclei) was estimated as follows:(1)$$\mathrm{tumor}\;\mathrm{size}=\;\frac{\mathrm{AVG}\;\left(nº\mathrm{of}\;\mathrm{DAPI}\;\mathrm{cells}\;\mathrm{Zfirst}\;+\;nº\mathrm{of}\;\mathrm{DAPI}\;\mathrm{cells}\;\mathrm{Zmiddle}\;+\;nº\mathrm{of}\;\mathrm{DAPI}\;\mathrm{cells}\;\mathrm{Zlast}\right)\;\times \;\mathrm{total}\;\mathrm{number}\;\mathrm{of}\;\mathrm{slices}}{1.5}$$

The number of mitotic figures and activated caspase-3 were quantified manually, counting all cells in every slice (from Zfirst to Zlast), and the respective percentages were generated by dividing the values by the tumor size (nº of tumor cells) of the respective tumor.

### 2.11. Histopathology

Zebrafish xenografts were fixed in 4% paraformaldehyde, paraffin-embedded, sectioned at 4 μm, and stained with hematoxylin and eosin for routine histopathological analysis. Microphotographs were captured in a Leica DM2000 microscope coupled to a Leica MC170 camera.

### 2.12. Statistical Analysis

Statistical analysis was performed using GraphPad Prism 8.0 software. All datasets were challenged by normality tests (D’Agostino and Pearson, and the Shapiro–Wilk). A Gaussian distribution was only assumed for datasets that passed both normality tests and were analyzed by an unpaired *t*-test with Welch’s correction. Datasets without Gaussian distribution were analyzed by unpaired and nonparametric Man–Whitney test. For all the statistical analyses, *p* value (*p*) is from a two-tailed test with a confidence interval of 95%. Statistical differences were considered significant whenever *p* < 0.05, and statistical output was represented by stars as follows: non-significant (ns) > 0.05, * *p* ≤ 0.05, ** *p* ≤ 0.01, *** *p* ≤ 0.001, and **** *p* ≤ 0.0001. All graphs present the results as average (AVG) ± standard error of the mean (SEM).

## 3. Results

### 3.1. Characterization and Histomorphological Features of Panc-1 Zebrafish Xenografts

We started by selecting a human PC cell line, Panc-1, which has been reported as highly aggressive and less differentiated than others [30], being commonly used to study pancreatic carcinogenesis.

Panc-1 cells were fluorescently labeled with CM-DiI and injected into the perivitelline space (PVS) of 2 days postfertilization (dpf) zebrafish embryos. To avoid capillary clogging due to the grape-like morphology of the Panc-1 cell line, tumor cells were resuspended in 1XPBS 2 mM EDTA before injection. At 1 dpi, zebrafish xenografts were screened to select the successfully injected xenografts and sacrifice the badly injected ones or those with severe edema [19]. Unfortunately, 73.38% of Panc-1 xenografts presented severe cardiac edema at this timepoint (Figure 1a). Several studies have shown that PC cells can trigger inflammation by producing and secreting proinflammatory cytokines, growth factors, and metalloproteases essential for PC progression [31,32]. In this way, this pathological condition displayed by zebrafish xenografts may be correlated with proinflammatory signals released by necrotic Panc-1 cells. From the ~30% that did not present edema at 1 dpi (Figure 1b), ~52% of these xenografts presented a tumor mass at 4 dpi, which we define as engraftment (Figure 1c). These tumors showed blood vessel recruitment in their base but not infiltrating into the tumor core; therefore, we considered these tumors as poorly angiogenic (Figure 1d).

In order to characterize the morphological features and localization of these tumors, Panc-1 xenografts were processed for histopathology at 4 dpi. We found that Panc-1 solid tumor masses were often in close proximity with the zebrafish pancreas, liver, and gut (Figure 1g–j’).

Finally, to analyze the capacity of these Panc-1 cells to proliferate in the zebrafish host, we quantified the mitotic index (mitotic figures and phosphohistone H3). Our results show that, although Panc-1 cells do not engraft very efficiently, they are able to actively proliferate in zebrafish xenografts (Figure 1k–n).

### 3.2. Pancreatic Cancer Zebrafish Xenografts Show Sensitivity to Standard Chemotherapy

To test whether zebrafish xenografts could be used to measure different responses to the main therapeutic options in advanced PC guidelines, we first determined the maximum tolerated concentration for these therapies. Based on the maximum plasma concentration found in patients [23,24,25,26,27,28] and on our previous experience [16], we tested three different concentrations: FOLFIRINOX#1 (8.4 mM 5-FU + 0.36 mM folinic acid + 0.16 mM oxaliplatin + 0.16 mM irinotecan); FOLFIRINOX#2 (6.3 mM 5-FU + 0.27 mM folinic acid + 0.12 mM oxaliplatin + 0.12 mM irinotecan); FOLFIRINOX#3 (4.2 mM 5-FU + 0.18 mM folinic acid + 0.08 mM oxaliplatin + 0.08 mM irinotecan); GnP#1 (3.2 mM gemcitabine + 730 ng/mL paclitaxel); GnP #2 (2.4 mM gemcibatine + 547.5 ng/mL paclitaxel); GnP#3 (1.6 mM gemcitabine + 365 ng/mL paclitaxel). We chose the maximum concentration that did not kill the fish or cause any visible physical disability (Figure 2).

Next, Panc-1 zebrafish xenografts were generated and randomly distributed between treatment groups (control, FOLFIRINOX, and GnP) at 1 dpi (Figure 3a–c). After three days of treatment (corresponding to 4 dpi), xenografts were processed for confocal microscopy and assessed for mitotic index, cell death by apoptosis (activated caspase-3), and tumor size (Figure 3d–f).

In Panc-1 tumors, both FOLFIRINOX and GnP regimens induced a significant reduction in mitotic figures (FOLFIRINOX: ~42% reduction, ** *p* = 0.0042; GnP: ~51% reduction, *** *p* = 0.0006; Figure 3d). A significant induction of apoptosis was also observed with both treatments (**** *p* < 0.0001; Figure 3a’–c’,e). FOLFIRINOX treatment significantly reduced the tumor size of Panc-1 tumors (FOLFIRINOX: ~24% tumor shrinkage, ** *p* = 0.0097). However, despite the cytotoxic effects regarding apoptosis induction and proliferation blockage, GnP only showed a tendency to decrease tumor size (GnP: ~18% tumor shrinkage; *p* = 0.0661) (Figure 3f). Overall, our results clearly show that we can detect the cytostatic and cytotoxic effects of the main therapeutic options for pancreatic cancer.

### 3.3. Zebrafish Avatars Can Be Generated from Human Pancreatobiliary Tumors of Different Histotypes

To test the feasibility of the zAvatar model in tumors developed in the pancreatic region, we used surgical pancreatectomy-resected samples without in vitro expansion to generate zebrafish Avatars from eight patients (Table 1, Figure 4). From these eight samples, five were PDACs, two were ampulla adenocarcinomas (AAC), one intestinal type and one biliopancreatic type, and, also, one distal cholangiocarcinoma (Table 1). In terms of tumor grade, samples ranged from poorly (G3), moderately (G2), and well differentiated (G1) (Table 1). All patients were treated at the Champalimaud Clinical Centre (CCC) and participated in this study after signed informed consent.

Tumor tissue was dissociated, and cell suspensions were fluorescently labelled for injection (see Methods). At 1 dpi, zAvatars were screened and subjected to treatment. In two samples, it was not possible to extract cells from the surgical specimen, probably due to low cellularity and excessive stroma (Table 1, Patient #7, #8). Two samples (zAvatar #3 and #6) induced a severe edema at 1 dpi, which persisted until 3 dpi (Appendix A), and therefore the engraftment was very low (only xenografts without edema were considered) (Figure 4b). Interestingly, similarly to the Panc-1 cell line, these two zAvatars induced the formation of severe edemas (Figure 1a).

The remaining four samples proceeded to chemotherapy treatment, according to the patient’s adjuvant regimen (when it was performed). Interestingly, these samples were composed of two PDAC and two AAC (Table 1), showing an implantation rate between 44–64% (Figure 4b) and tumor sizes ranging from ~30 to 200 cells per xenograft at 3 dpi (Figure 4d).

To determine the sensitivity or resistance to treatment, apoptosis induction and tumor size were analyzed as previously (Figure 4c,d). We could detect a significant impact on apoptosis in zAvatar#1 and zAvatar#4, with a fold induction of activated caspase-3 of 1.5 (*p* = 0.001) and 2.12 (*p* = 0.0061), respectively (Figure 4c–l). However, we could only observe a significant shrinkage of the tumors in zAvatar#1 (Figure 4d, *p* = 0.05). In contrast, we could not detect any significant change in apoptosis or tumor size in zAvatar#2 and #5 (Figure 4c–l), suggesting that these tumors were resistant to the tested treatment options.

Of note, we could observe recruitment of endothelial cells in zAvatars #1, #2, and #4, suggesting that these tumors have a high angiogenic potential (Appendix A).

It was not possible to make a correlation between the response of the zAvatars and the patient clinical response, since one patient was still under chemotherapy (#5), one did not go for adjuvant treatment (#1), and the other two (#2 and #4) still have a short follow-up to evaluate clinical response. Nonetheless, we optimized the pancreatic protocol and showed the possibility to study different treatment approaches.

Finally, we performed a quality control examination of the injected samples, by analyzing the cytomorphological features of the tumor cells before injection in May-Grunwald–Giemsa-stained smears (Appendix A). We found that samples that induced severe edema (#3 and #6) in the zAvatars were rich in poorly cohesive cells, with necrosis and abundant apoptotic debris. Moreover, the presence of bacteria was detected in sample #6, possibly explaining its poor/null engraftment. On the other hand, the smears of the successfully injected samples were composed of highly cohesive cell aggregates, with few nuclear tangles, minimal necrosis, and low apoptotic debris (Appendix A).

In summary, here we demonstrate that it is possible to establish zebrafish Avatars from human pancreatic and ampullary cancers and detect differential responses to the standard therapies with single cell resolution.

## 4. Discussion

In the present study, we provide a short report where we tested the feasibility and sensitivity of the zebrafish xenograft model to screen the main therapeutic options for pancreatobiliary cancers.

First, we selected a human PC cell line, Panc-1, to optimize all protocols and study pancreatic cancer cell behavior in the zebrafish host. One striking characteristic was the high incidence of severe cardiac/abdominal edema at 1 dpi. Although in other cancer models we also observe cardiac edemas due to injection, these were never with such high incidence and so early in the assay (1 dpi). This severe edema may be related to the reported overexpression of proinflammatory cytokines (IL-1, IL-6, IL-8, and TNFα) and matrix metalloproteases [33,34] secreted by PC cells [32], and may provide an interesting model to explore the formation of edema.

In order to test whether zebrafish xenografts could discriminate different sensitivities to the available therapies approved for advanced PC, Panc-1 zebrafish xenografts were subjected to FOLFIRINOX and GnP therapies. After three consecutive days of treatment, we were able to detect that both antineoplastic drugs significantly impaired the number of cancer cells underdoing mitosis and significantly induced apoptosis. FOLIFIRINOX treatment promoted a significant shrinkage of the tumor mass, while GnP showed only a tendency to decrease tumor size. Our results showing the sensitivity of Panc-1 cells to FOLFIRINOX and GnP are in agreement with previous in vitro studies [35,36,37,38,39,40,41].

The establishment of zAvatars from pancreatic surgical samples is challenging due to the low cellularity of the samples and the abundance of a desmoplastic dense stroma. Nevertheless, we were able to establish a zAvatar model from different types of tumors developed in the pancreatic region (PDACs and AAC), treated with different chemotherapies, and analyzed through confocal microscopy with single-cell resolution. We were not able to perform correlations with the patient’s response due to the short follow--up and reduced number of patients, but we were able to set the groundwork for a future clinical study to test the predictiveness of the model.

As a comment, we and others have shown that zebrafish avatars are a promising model to predict response in patients of chemo- [16,42,43,44,45] and radiotherapy [17], but also targeted therapies, such as anti-EGFR (cetuximab) [16], anti-VEGF (bevacizumab) [18], and PARPi (olaparib) [20]. As for immunotherapies, the model is more limited. The larval xenograft assay is performed in a developmental stage where only innate immunity is at play; adaptive immunity is still not matured. Therefore, in principle it is only possible to access response to immunotherapies in which the underlying driver of therapeutic efficacy is innate immunity, and not adaptive immunity.

Some studies have been published in PC using the zebrafish model [42,43,44,45], but, to the best of our knowledge, only one challenged patient-derived cells to the current major therapeutic options according to guidelines [45]. Di Franco et al. 2020 [45] evaluated the use of zebrafish xenografts as Avatars for PC patients. Similarly, to our report, Di Franco et al. 2020 showed sensitivity to both FOLFIRINOX and GnP schemes, but no correlation with patient outcome was performed, also due to short follow-up. However, it is important to highlight some differences in the experimental design/techniques and read-outs of this study in relation to ours, such as: Di Franco et al. transplant small tissue fragments into the yolk, whereas we inject a cell suspension in the PVS; Di Franco et al. analysis was only based on lipophilic dye intensity area, whereas we analyze and show several readouts by single-cell confocal microscopy (tumor size, apoptosis, and human mitochondria).

Most studies using xenografts analyze the impact of treatment using unspecific methods, such as measuring tumor volume from fluorescence intensity of the lipophilic dye. Lipophilic dyes, although very useful, are not very reliable, since some cells stain better than others and dead cells or debris can be retained in the yolk sac, leading to false signals. Moreover, zebrafish phagocytic cells, such as macrophages, can become stained after “eating” these debris, leading also to false positive signals. Therefore, we use an anti-human mitochondria (hMITO) antibody as a quality control for our xenografts. hMITO labels exclusively human cells and serves as a marker to distinguish human cells from cellular debris or zebrafish phagocytes. This, together with DAPI staining and activated caspase-3 analysis, allow us to quantify induction of apoptosis and tumor size to evaluate treatment outcome in a more thorough manner. Moreover, we performed a quality analysis of the injected human samples, which is important to understand the heterogeneity of tumors and the results obtained considering the unique characteristics of each one (Appendix A).

Finally, it is important to mention that since most pancreatic patients present in advanced stages, either with metastasis or unresectable tumors, systemic chemotherapy is the only treatment option and echoendoscopic-ultrasound-guided biopsies are the only samples that will be collected. Thus, it is in this clinical setting that the knowledge of chemotherapy sensibility could have a critical clinical impact for patient management. Therefore, another challenge lies ahead: establishing zAvatars from pancreatic biopsies, which may yield even less cells.

## 5. Conclusions

Altogether, we provide a short report establishing the feasibility of generating and analyzing zAvatars from pancreatic and ampullary cancers by single-cell confocal imaging, with potential to be used for future preclinical studies and personalized treatment.

## Figures and Tables

**Figure 1 cells-10-02077-f001:**
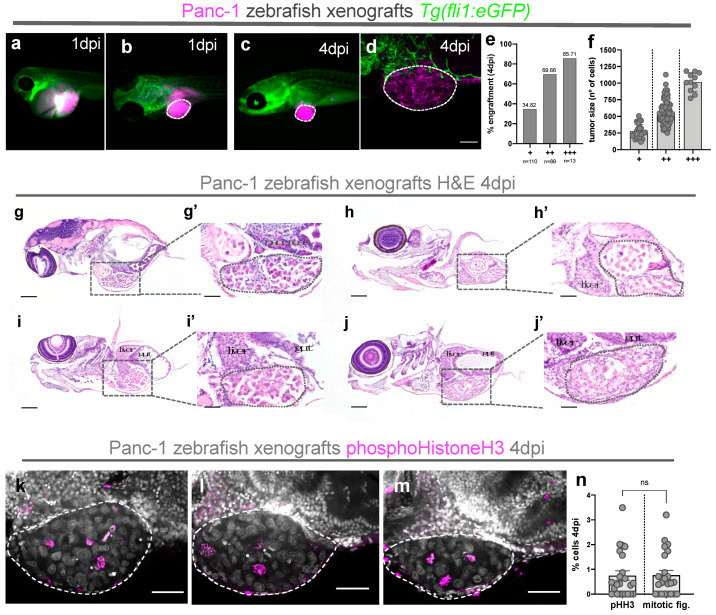
Characterization and histological analysis of human pancreatic cancer zebrafish xenografts. Human Panc-1 cells were fluorescently labeled with DiI (magenta) and injected into the perivitelline space (PVS) of 2 days postfertilization (dpf) Tg (*fli1:eGFP*) zebrafish embryos (**a**–**d**). At 1 day post-injection (dpi), zebrafish xenografts with severe cardiac edema were discarded (**a**), and successfully injected xenografts (**b**) were divided according to their tumor size (+, ++, +++) and were kept at 34 °C until the end of the experiment. At 4 dpi, zebrafish xenografts were evaluated regarding the presence of tumor (% engraftment) (**c,e**), tumor size (**f**), and angiogenic capacity (**d**). Representative H&E-stained microphotographs of paraffin-embedded Panc-1 zebrafish xenografts at 4 dpi showing how Panc-1 cells implant in close proximity to the zebrafish pancreas, liver, and gut (**g**–**j’**). Scale bars represent 100 µm. (**k**–**m**) Representative confocal images of mitosis revealed by pHH3 staining (magenta) and DAPI (labeling condensed chromatin that enables detection of mitotic figures) and quantification of mitotic index—% mitotic figures and % pHH3 (**n**). All results are expressed as AVG ± SEM and correspond to at least 3 independent experiments. The number of zebrafish xenografts analyzed is indicated, and each dot represents one zebrafish xenograft. Scale bars represent 50 µm. All images are anterior to the left, posterior to right, dorsal up, and ventral down.

**Figure 2 cells-10-02077-f002:**
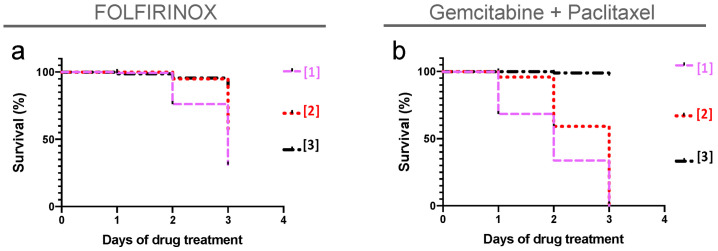
Maximum tolerated concentration (MTC) for FOLFIRINOX and gemcitabine + paclitaxel. Using as a reference the maximum patient’s plasma concentration of each compound (Appendix A), we determined the zebrafish MTC. Noninjected embryos with 3 dpf were treated and replaced daily with different doses of chemotherapy for three consecutive days. (**a**) FOLFIRINOX [1] corresponds to 8.4 mM 5-FU + 0.36 mM folinic acid + 0.16 mM oxaliplatin + 0.16 mM irinotecan; FOLFIRINOX [2] corresponds to 6.3 mM 5-FU + 0.27 mM folinic acid + 0.12 mM oxaliplatin + 0.12 mM irinotecan; FOLFIRINOX [3] corresponds to 4.2 mM 5-FU + 0.18 mM folinic acid + 0.08 mM oxaliplatin + 0.08 mM irinotecan. (**b**) GnP [1] corresponds to 3.2 mM gemcitabine + 730 ng/mL paclitaxel; GnP [2] corresponds to 2.4 mM gemcibatine + 547.5 ng/mL paclitaxel; GnP [3] corresponds to 1.6 mM gemcitabine + 365 ng/mL paclitaxel. Due to the reduced mortality, the concentration [3] depicted in black was chosen for both combinations.

**Figure 3 cells-10-02077-f003:**
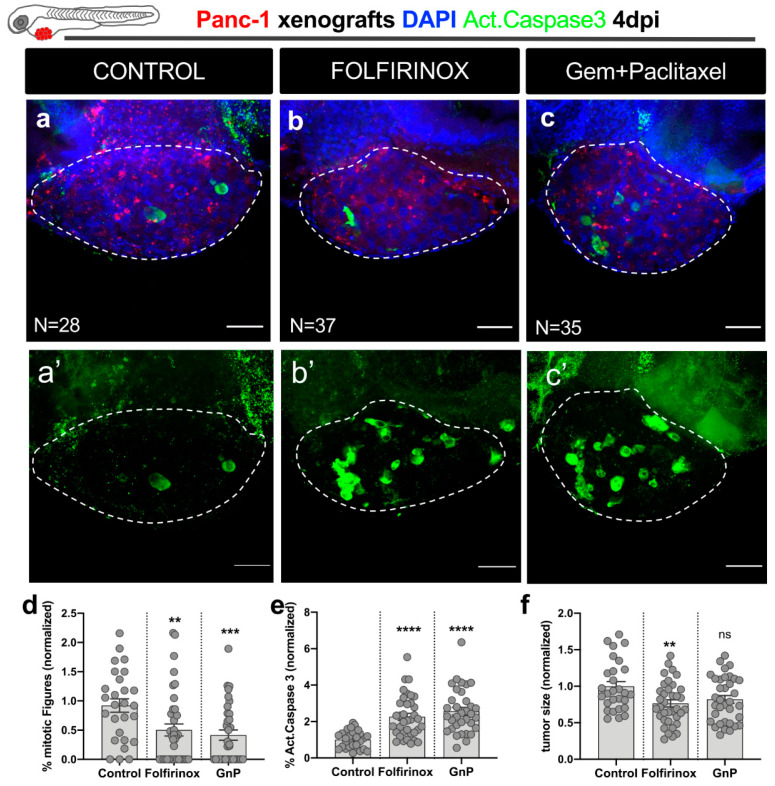
Zebrafish xenografts reveal sensitivity to the major therapeutic options for pancreatic cancer— FOLFIRINOX and gemcitabine plus paclitaxel chemotherapy. At 2 dpf, zebrafish embryos were injected with fluorescently labeled Panc-1 cells in the PVS. At 1 dpi, successfully injected xenografts were submitted to FOLFIRINOX or gem + paclitaxel treatment for three consecutive days and compared to control nontreated xenografts. At 4 dpi, zebrafish xenografts were imaged by confocal microscopy (**a**–**c**). Maximum Z projections of activated caspase-3 (**a’**–**c’**). Cell proliferation (% of mitotic figures), apoptotic index (% of activated caspase-3 in green), and tumor size (number of tumor cells, DAPI in blue) were analyzed and quantified (**d**–**f**, respectively). Data are shown as mean ± SEM. Statistical analysis was performed using Mann–Whitney test. Statistical results: (ns) > 0.05, ** *p* ≤ 0.01, *** *p* ≤ 0.001, **** *p* ≤ 0.0001. The number of zebrafish xenografts analyzed is indicated in the figure. Results are from 3 independent experiments. Scale bars represent 50 µm. All images are anterior to the left, posterior to right, dorsal up, and ventral down.

**Figure 4 cells-10-02077-f004:**
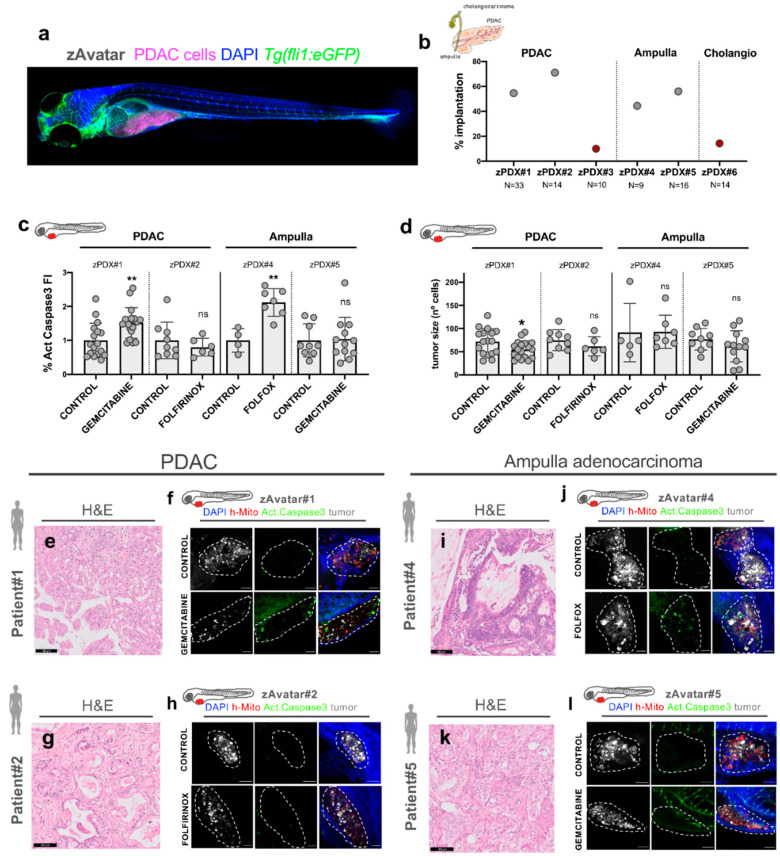
zAvatars generated from different histological pancreatobiliary tumors. For all examples, cells were extracted from the surgical resected sample, injected in 2dpf embryos, and tested for chemotherapy schemes. At 3 dpi and 2 days post-treatment (2 dpT), zAvatars were sacrificed and fixed, subjected to whole-mount immunofluorescence and imaged by confocal fluorescent microscopy. (**a**) Confocal image representative of a zAvatar injected with PDAC cells (labeled in magenta) in a Tg (*fli1:eGFP*) zebrafish background (labeled in green) at 3 dpi. (**b**) Percentage of engraftment of the different zAvatars; the total number of xenografts analyzed at the end of the assay is depicted in the figure. (**c**) Quantification of the apoptotic index (fold induction of activated caspase-3). (**d**) Quantification of the tumor size (fold change of the number of tumor cells). All results are expressed as AVG ± SEM. Statistical analysis was performed using Mann–Whitney test. Statistical results: (ns) > 0.05, * *p* ≤ 0.05, ** *p* ≤ 0.01. Each dot represents one zAvatar. (**e,g,i,k**) Hematoxylin and eosin staining of Patient #1, #2, #4, and #5 pathology samples, respectively. Scale bars represent 100 µm. (**f,h,j,l**) Representative confocal images of control and treatment zAvatars #1, #2, #4, and #5, respectively. Note that only zAvatar #5 is a Tg (*fli1:eGFP*), where zebrafish blood vessels can be detected in green. Scale bars represent 50 µm. For all images, tumor cells are labeled in white, activated caspase-3 in green, human mitochondria in red, and DAPI in blue. All images are anterior to the left, posterior to right, dorsal up, and ventral down.

**Table 1 cells-10-02077-t001:** Clinical characterization of patients included in the study.

Patient	Staging	Histologic Subtype	Tumor Size	R0	Patient CT Regimen
**#1**	T2N0	pancreas ductal adenocarcinoma with areas of papillary cystic G2	38 mm	yes	0
**#2**	T3N0	pancreas ductal adenocarcinoma G2	56 mm	yes	FOLFIRINOX
**#3**	T2N0	pancreas ductal adenocarcinoma G1 (associated to IPMN)	23 mm	no	FOLFIRINOX
**#4**	T3N1	ampulla adenocarcinoma, intestinal type (mucinous 40%) G2	45 mm	yes	FOLFOX
**#5**	T3N0	ampulla adenocarcinoma, biliopancreatic type G2	15 mm	yes	Gemcitabine
**#6**	T3N1	distal cholangiocarcinoma G2	17 mm	yes	0
**#7**	T2N1	pancreas ductal adenocarcinoma G2	34 mm	yes	0
**#8**	T3N2M1	pancreas ductal adenocarcinoma G3	48 mm	yes	FOLFIRINOX

## Data Availability

The data presented in this study are available on request from the corresponding author.

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
