# Peer review of "Establishment of Pancreatobiliary Cancer Zebrafish Avatars for Chemotherapy Screening"

_cells, 2021, doi:10.3390/cells10082077_

Round 1

Reviewer 1 Report

The manuscript entitled “Establishment of pancreatobilary cancer zebrafish avatars for chemotherapy screening” by Mariana Tavares Barroso and coworkers is a brief report on the use of xenotransplantation in zebrafish embryos as a way to characterize pancreatic-derived cancer cells and their response to therapeutic agents. The report concerns one cell line, Panc-1, as well as tumor materials derived from 8 patients.

The manuscript is well written, presented in a comprehensive fashion and clear. The methods are nicely detailed. The experiments are well conducted and the conclusions supported by the data presented. However, I believe that the manuscript could be improved if the authors would address, specify or comment the following points:

1.The authors are using two zebrafish lines, Nacre and Tg(fli1:GFP) as recipients in the transplantation experiments. However, it is not always clear which one is used in a given experiment. For instance, Fig 4a clearly shows a Tg(fli1:GFP) embryos, but what is about Fig. 4f,j,h,l? Does the green signal outside of the tumor corresponds fli1:GFP cells or to activated caspase 3 signals. This needs to be clarified.

2. As stated in the Discussion section, the use of the anti-human mitochondrial antibody contributes to quality control of the xenograts. Could the authors indicate the precise reference of this antibody in the Materials and Methodes section? References of the other antibodies used should also be specified.

3. In the experiments using Panc-1, the effects of drug treatments are investigated at 4 dpi (3 dpT), but when patient-derived cells are used, the experiments are stopped at 3 dpi (2 dpT). What is the rational of this difference in the experimental procedure? Is this difference (2 dpT versus 3 dpT) impacting the responses to the treatments?

4. Concerning the engraftment of the different zAvatars (Fig. 4b), could the authors indicate the number of xenografts performed to obtain the values of % implantation in the Figure?

5. In the Introduction section, FOLFOX does not appear as a treatment for ampulla adenocarcinomas. However, FOLFOX regiment is applied to Patient #4 and used in the corresponding zPDX. Even if the clinical response to Patient #4 is not known an even if FOLFOX has not been tested on Pac-1 transplanted cells in the manuscript, could the authors discuss the use of FOLFOX as a possible therapy in addition to Gencitabine for ampulla adenocarcinomas in their introduction.

Minor points:

1. For better clarity, could the authors explicitly indicate the drug concentrations used in the body of the text page 7, line 232 (even if it is already indicated in the Fig 2 legend and in the Materials and Methods)?

2. Typo mistakes are present in the reference list. Numerous references (starting at Ref. 11) have twice the number.

Author Response

We would like to thank reviewer#1 for the critical and careful reading of our manuscript and the opportunity to address all concerns raised, improving our manuscript

Please see attached file with response point by point.

Reviewer 2 Report

In this article Barosso and colleagues bring forward a zebra fish model of pancreatic cancer with potential to predict the outcome to chemo therapy in shorter time. The authors used Panc1 cell line derived and patient derived tumors that are injected in the perivitelline space of zebra fish. An indepth characterization of tumor vasculature and microenvironment is provided. Predictions and response towards commonly used chemo FOLFIRINOX and gemcitabine is also presented.

Overall the studies are clearly presented and add to the field of pancreatic cancer models.

Minor comments:

Discussion on the implications of this model on predicting outcomes of targeted and immunotherapies should be included.

Author Response

We would like to thank reviewer#2 for the critical and careful reading of our manuscript and the opportunity to address concerns raised, improving our manuscript.

In this article Barosso and colleagues bring forward a zebra fish model of pancreatic cancer with potential to predict the outcome to chemo therapy in shorter time. The authors used Panc1 cell line derived and patient derived tumors that are injected in the perivitelline space of zebra fish. An indepth characterization of tumor vasculature and microenvironment is provided. Predictions and response towards commonly used chemo FOLFIRINOX and gemcitabine is also presented.

Overall the studies are clearly presented and add to the field of pancreatic cancer models.

Minor comments:

Discussion on the implications of this model on predicting outcomes of targeted and immunotherapies should be included.

We thank reviewer#2 for the suggestion, we have included a small comment in the discussion- see below:

As a comment we and others have shown that zebrafish Avatars are a promising model to predict response in patients of chemo [16, 42-45], radio [17] but also targeted therapies such as anti-EGFR (cetuximab) [16], anti-VEGF (bevacizumab) [18] and PARPi (Olaparib) [20]. As for immunotherapies the model is more limited. The larval xenograft assay is performed in a developmental stage where only innate immunity is at play, adaptive immunity is still not matured. Therefore, in principle, it is only possible to access response to immunotherapies in which the underlying driver of therapeutic efficacy is innat­­­e immunity and not adaptive immunity.

Round 2

Reviewer 1 Report

I thank the authors for their responses to my comments, and wish to congratulate them for their nice study on the use of zebrafish avatars against prancreatic cancers.